# Applicability Domain of Active Learning in Chemical Probe Identification: Convergence in Learning from Non-Specific Compounds and Decision Rule Clarification

**DOI:** 10.3390/molecules24152716

**Published:** 2019-07-26

**Authors:** Ahsan Habib Polash, Takumi Nakano, Shunichi Takeda, J.B. Brown

**Affiliations:** 1Kyoto University Graduate School of Medicine, Department of Molecular Biosciences, Life Science Informatics Research Unit, Kyoto, Sakyo, Yoshida, Konoemachi, Kyoto 606-8501, Japan; 2Kyoto University Graduate School of Medicine, Department of Radiation Genetics; Kyoto, Sakyo, Yoshida, Konoemachi, Kyoto 606-8501, Japan

**Keywords:** chemical probes, compound specificity, ligand-target interactions, chemogenomics, active learning, active projection, decision tree, molecular representation

## Abstract

Efficient identification of chemical probes for the manipulation and understanding of biological systems demands specificity for target proteins. Computational means to optimize candidate compound selection for experimental selectivity evaluation are being sought. The active learning virtual screening method has demonstrated the ability to efficiently converge on predictive models with reduced datasets, though its applicability domain to probe identification has yet to be determined. In this article, we challenge active learning’s ability to predict inhibitory bioactivity profiles of selective compounds when learning from chemogenomic features found in non-selective ligand-target pairs. Comparison of controls versus multiple molecule representations de-convolutes factors contributing to predictive capability. Experiments using the matrix metalloproteinase family demonstrate maximum probe bioactivity prediction achieved from only approximately 20% of non-probe bioactivity; this data volume is consistent with prior chemogenomic active learning studies despite the increased difficulty from chemical biology experimental settings used here. Feature weight analyses are combined with a custom visualization to unambiguously detail how active learning arrives at classification decisions, yielding clarified expectations for chemogenomic modeling. The results influence tactical decisions for computational probe design and discovery.

## 1. Introduction

Following the sequencing of the human genome, the term chemogenomics emerged to represent the study of how chemicals interact with the protein products of a genome [1]. Chemogenomics, which combines the concepts that similar ligands bind to similar targets and that similar targets bind similar ligands, thus bears an obvious yet crucial relationship to the field of chemical biology, where compounds with high specificity for a particular target are sought, in addition to the base requirement of potency [2]. Such specificity improves the ability to unambiguously modulate a biological system toward elucidation of state/phenotype-target relationships, often referred to as phenotypical target deconvolution [3].

Computational chemogenomics and computational chemical biology are then concerned with the application of computational methods as a means of achieving the goals of their respective experimental disciplines. As a full ligand-target matrix of the entire chemical universe screened against the reference human proteome is infeasible, computational chemogenomics is attractive for its ability to infer untested ligand-target interactions from the similarity of existing ligand-target interaction data [4,5,6], and as a corollary, computational chemogenomics provides a means for achieving goals in computational chemical biology (e.g., computational analysis of potency and selectivity). By the development of computational methods, the prioritization of experimental evaluations of ligand-target interactions can be achieved.

Chemogenomic virtual screening of ligand-target interactions has been studied for many years, where early studies [7,8,9] focused on the design and use of kernel methods [10] to satisfy mathematical requirements needed for guaranteed optimal solutions by the Support Vector algorithm [11]. It is commonly thought that increasingly larger ligand-target datasets result in increasingly better chemogenomic models, though computation of kernel-adapted estimators increases quadratically with the number of examples to fit models to; recent studies have largely turned attention to artificial neural networks (ANNs), and in particular deep neural networks (DNNs), that support small batch learning and update [12,13,14]. Yet ANNs and DNNs, typically built by iterative backpropagation of prediction errors where gradient descent methods update decision parameters toward stable prediction convergence (minimized error) in each node in the network topology, do not contain mathematical guarantees on optimality as support vector models do, nor are there biophysical or pharmacological premises that justify expectations for very large volumes of data to yield better estimators.

A few years ago, Reker et al. demonstrated that chemogenomic data of large protein families such as GPCRs and kinases could be dynamically analyzed in such a way that systematic selection of 5–25% of the bioactivity data was sufficient to predict the entire collection of ligand-target interactions with high performance [15]. This method, chemogenomic active learning, was subsequently shown to also be predictive on smaller target families with highly sparse annotation and further assessed for its domain of applicability by prediction on individual target bioactivity profiles withheld from available training data [16]. In both studies, the underlying model technique used was the Random Forest [17], an alternative to the SVM and DNN model fitting methods, which is also capable of detecting non-linear relationships.

More generally, active learning platforms are composed of two components, namely an estimator (model) computation component (e.g., RF/SVM/ANN) and a data selection component that uses the results of a fitted estimator in order to pick an instance (or instances) from the available data which will then be included in subsequent fit-predict-pick cycles. Optionally, an external prediction set can be evaluated at each model update, as done previously [16] and in this work. Examples to include in the initial estimator fit could be randomly or intentionally selected (e.g., one ligand-target interaction and one non-interaction).

The strategy to iteratively pick examples by model feedback plays a dominant influence in the evolution of the prediction model. Random selection of instances regardless of predictions is used as a control experiment and is influenced by dataset ratio. Hence, in screening libraries where hits are infrequent [9,18,19,20], random selection will lead to a model that tends toward classification of all data as inactive. To recover and incorporate the minority actives into training, the greedy/exploitative selection strategy picks the instance which receives the most prediction expectation; RF-based estimators, fit to “0/1”-labeled data, “greedily” pick the ligand-target pair which was maximally classified as active (“1”) by decision trees comprising the forest. In contrast, the curiosity/explorative selection strategy picks the instance which has the most uncertainty in prediction (i.e., no definitive decision toward active or inactive classification, making the instance particularly “curious”); RF-based estimators choose the example with maximum variance in decision tree predictions. As a consequence, bioactivity examples positioned on the boundaries of active-inactive spaces are systematically uncovered; this is, not coincidentally, similar to the idea of support vectors in the SVM algorithm. For chemogenomic studies, the curiosity picker typically displays early convergence on balanced active-inactive selection and rapid gains in prediction performance. In the case of single-target or phenotypical bioactivity endpoint modeling, both the greedy and curiosity picking strategies are more effective than random picking, assuming data is not biased toward actives.

While the biological endpoint withholding and prediction study [16] clarified active learning’s domain of applicability from biological and pharmacological points of view, the compounds included in the chemogenomic data were filtered only to remove any inconsistencies in activity annotations. Thus, while potency was considered, the impact of compound promiscuity on modeling results, and therefore, on chemogenomic active learning’s domain of applicability in chemical biology, has yet to be fully explored. Addressing this issue would clarify the extent to which active learning can rapidly learn rules applicable to compound design bearing specificity in mind.

In this article, we examine the active learning methodology with a particularly challenging task of predicting the ligand-target interaction profiles of compound probes that demonstrate potency and selectivity for a target. Importantly, the challenge is designed such that chemogenomic active learning can only learn from patterns in compounds that are not probes (i.e., not potent on a target, not selective for a target, or both). To understand exactly how a model dynamically evolves during the process of chemogenomic active learning, a recent technique for multi-metric time-series tracing of predictive ability is combined with dynamic feature weight tracing and customized decision tree rendering, in addition to a standard evaluation of longitudinal prediction performance. We uncover that active learning can indeed be predictive on the profiles of probe compounds that are external to the collection of non-probe ligand-target bioactivities, and that target descriptors do in fact factor into and boost the decision making of decision trees comprising a chemogenomic random forest model. By examining rendered decision trees in depth, we arrive at a conclusion that highly-performant chemogenomic modeling is achieved by the creation of many local SAR models within the global model, suggesting that selection between chemogenomic and single-target models for probe development is determined by the availability of data for the target to be modulated.

## 2. Results and Discussion

### 2.1. Generation of Probe and Non-Probe Datasets

The process of data retrieval, discretization, and split into non-probe training and probe external prediction datasets is given in Figure 1. Inhibitory activities against the matrix metalloproteinase (MMP) family, a collection of more than 20 enzymes with both tumor suppression and tumor progression roles [21], were retrieved from ChEMBL [22]. MMP inhibitors were once expected to be a central strategy for inhibition of cancer growth, yet promising preclinical developments in small molecules did not translate to successful clinical trials [23]. Despite the setback, there is renewed interest in development of small molecules for MMPs due to their roles aside from oncology [24], and antibodies for several MMPs are commercially available.

MMP targets with sufficient *K_i_* measurements were used to derive active and inactive annotations, with ligands subsequently categorized into probe and non-probe statuses (see Methods). In short, probes had strong potency for exactly one MMP target and non-potency for at least one other MMP. The non-probe dataset was made available to the active learning method for training, and the external probe dataset was tested for predictability at each iteration of the fit-predict-pick active learning cycle. This choice of train-test data represents a chemical-centric domain of applicability challenge different from a prior target-centric study [16].

Table 1 shows the statistics of the resulting dataset. Nine MMP targets had sufficient ligand bioactivity data to be included in the study. In total, there were, respectively, 1181 and 72 unique compounds among 2397 and 165 bioactivities in the training and external datasets. 750 compounds in the non-probe training data contained at least one active annotation; among them, 708 (94%) had exclusively active annotations. 473 compounds contained at least one inactive annotation; among them, 431 (91%) had exclusively inactive annotations. 42 compounds were present with both inactive and active annotations. For the external set of probe bioactivities, in accordance with definition, all 72 compounds have one active annotation and at least one inactive annotation (c.f., Figure 1).

No compound satisfying the requirements to be included as a probe contains annotations against MMP12; all ligands with MMP12 annotation are present only in the non-probe training bioactivity data. The percentages of actives per target in the training data ranged from 19% (MMP7) to 88% (MMP12). With respect to fraction of active annotations per target between the training and test sets, the overall trend was positive (Pearson R 0.73); MMP1 did not follow the trend, with 54% (200) active annotations in the training data but only 4% (2) active annotations in the test data.

### 2.2. Longitudinal Performance of Active Learning Strategies and Descriptor Impact

The first probe profile prediction challenge executed was a control experiment for which subsequent experiments could be gauged for significance. We chose simple physicochemical descriptions of compounds (molecular weight, total polarizable surface area, estimated LogP, etc.) and combined them with a dummy identity descriptor for each target (see Methods). That is, each MMP target has a unique descriptor profile with a single value of unity in one position and a value of zero in all other positions. Combining all identity descriptor profiles results in an identity matrix. By using this descriptor and then comparing results to those obtained from a more biologically relevant descriptor, the contribution of a biological descriptor can be assessed more objectively.

Using the control descriptor, an initial active learning of and prediction on the training (non-probe) data, which can be construed as retrospective active learning, was done with evaluation of each of the random, greedy, and curiosity strategies (see Introduction). The F1 measure, a metric combining correct active predictions, type-I error, and type-II error, as well as the Matthews Correlation Coefficient, a metric incorporating all types of prediction results with multiplicative penalty for mistakes and thus reflects accounting for specificity, were chosen as evaluation metrics (see Methods for formulas). Compared to the well-known Accuracy metric (fraction of correct predictions) or recently proposed Power Metric [25], these two metrics are less subject to over-estimating model performance [26].

The baseline evaluation with random picking resulted in an average (*n* = 10) Matthews Correlation Coefficient (MCC) value of 0.70 (see Methods for metric formulas) at the point by which 20% of the available training data had been selected. At the same data volume, the greedy picker had an MCC of 0.30, whereas the curiosity picker had an MCC of 0.90 (Figure 2a). At 80% of the training data, all picking methods yielded models with MCC values greater than 0.90, meaning that the structure-activity relationships (SAR) of the remaining 20% of data could be inferred from the 80% used by any method.

The more crucial question and second experiment, prospective active prediction of the external probe dataset, was performed. Whereas in retrospective or recall-type active learning with the physicochemical-identity descriptor the curiosity-picked model achieved MCC of 0.90 with 20% of the training data, the same model was not predictive of the probe compound bioactivity profiles. In fact, none of the physicochemical-identity SARs developed by the picking methods were predictive against the challenge (MCC ~0.10, Figure 2a). At best, the curious picker at 35% of the data reached a peak of MCC = 0.20 and then dropped to MCC = 0.18 after the addition of more SAR training data. The other picking strategies arrived at the same predictive performance at 80% of the training data.

We proceeded to execute the same challenge but to describe the MMP targets in a more biologically relevant way and ask if SAR modeling would be improved. Identity description of MMPs was replaced with their dipeptide frequencies as protein descriptors (see Methods for calculation details). This did not make any mentionable changes in the retrospective model evaluation; prediction performance for each selection strategy was similar to that achieved using the identity descriptor. However, for the external dataset, improvement was marked. The curious picker reached MCC of 0.46 with 20% of the training data, which further improved to MCC = 0.50 at 23% (Figure 2b). Interestingly, performance decayed after including more than 30% of the training data, with stable convergence to MCC = 0.45 at 45% of the training data. The random and greedy pickers obtained respective MCC values of 0.30 and 0.10 at the benchmark 20% training data point, and eventually converged to the same MCC = 0.45 at later stages.

Intrigued by the prediction improvement on the probe data via the inclusion of dipeptide descriptors, we sought to clarify the contribution of the protein sequence information. Experiments were repeated with single amino acid frequency and tripeptide frequency descriptors, keeping the compound physicochemical description constant. Separately, we wanted to know the extent by which alternative chemical descriptions would affect prediction performance, and if the improvement over the identity target descriptor could be reproduced with a different chemical perception. In total, eleven additional experiments analogous to Figure 2 were executed, with all 13 experiments summarized in Table 2 and discussed next.

Exchanging the control target protein identity descriptor by amino acid, dipeptide, and tripeptide frequencies led to a continuous improvement in probe profile MCC performance, with respective MCCs of 0.35, 0.46, and 0.48 at 20% of non-probe training data (all experiments curiosity picking). Despite the fact that sequentially neighboring residues are not necessarily spatially near, there is a clear difference that addition of amino acid and peptide frequencies boosted the prediction performance of the model on the external dataset. As the underlying estimator algorithm (random forest) is not designed with biological or chemical context in mind, the best we could initially reason is that the extra protein description yielded increased matching correlation with distributions in the physicochemical description, that decision trees subsequently could identify these differential distributions, and that rules based on such differential distributions were transferrable to the external probe SAR data in many cases.

Dipeptide and tripeptide peak performance on the probe dataset was achieved at approximately 32% of training data, with MCC values above 0.52 (Table 2). In these experiments as well, curious active learning outperformed random and greedy strategies (Appendix A). The results clearly re-iterate that the assumption of more data yielding better models is invalid.

Next, we address the latter of two questions posed above by changing the perception of chemical space from physicochemical properties to structural fingerprints. The first descriptor change was to use atomic neighborhood fingerprints (ECFPs). A clear improvement in prospective prediction performance was again observed when replacing identity target descriptors with dipeptide descriptors (see Appendix A and Table 2). In the case of ECFPr2-1024 (maximum neighborhood radius of 2 from a central atom, with fingerprints hashed into 1024 bits), the change to a biological descriptor improved average performance on the external dataset (at 20% training) from MCC of 0.23 to 0.59. Additionally, the peak of mean MCC of the same system was 0.67 at 57% of training data.

Given these results from separate physicochemical and structural perceptions of compounds, we sought to know whether combining the two different types of descriptors could improve predictive performance. Upon combining the pChem and ECFPr2-1024 descriptors with dipeptide description, the curiousity picker scored a mean MCC of 0.57 at 20% of the training data, with a surprising peak mean MCC of 0.62 at 21% of the data (Table 2 and Appendix A). In this situation there was thus no gain from joining complementary chemical space descriptions.

To address the question if compression to more or less hash bits in the ECFP representation of compounds had any effect on external performance, additional experiments were undertaken. Quite unexpectedly, a change from 1024 bits to either 512 or 4096 bits resulted in improvement (Table 2 and Appendix A). While the expansion to more bits to reduce hashing collisions and empower the decision trees to find additional separation criteria appears rational, the details of how fewer bits could yield improved average performance is unclear and will require further investigation. The 4096-bit representation was notable in that prospective active learning performance was generally stable after achieving peak performance, unlike most other representations.

Finally, we additionally considered a pharmacophoric representation of compounds. The combination of CATS2D chemical descriptors (see Methods) and dipeptide frequency also resulted in improvement over the physicochemical description, and once again there was marked improvement when replacing the target identity descriptor with the dipeptide descriptor. Extension to tripeptide sequences led to a minor (+ ~0.03 MCC) yet improvement. Further testing with tetrapeptide subsequences showed no major change in long-term performance though marginal gains were made in early (5–15%) stages.

One remarkable feature for all external predictions using the curious picking strategy was that model performance peaked after learning from 20–25% of the training data. Further learning events yielded performances that were either unchanged or degraded. While performance in prior experiments [15,16,27,28] suggested that prediction performance monotonically increases with more data, those results were all in retrospective contexts; indeed, Rakers and colleagues [16] found that in biological contexts of simulated de-orphanization, performance could recede with increases in training data. Here, this finding has been replicated in a chemical context.

In addition to the MCC, it is equally prudent to consider the F1 score, which places emphasis on the ability to detect active (inhibiting) annotations. Values in Table 2, notably peak F1 scores, suggest that the majority of probe compounds could have their inhibiting annotation detected; corresponding MCC values were a bit lower, with the difference in metric values signaling that specificity (True Negative) prediction was the challenging aspect.

### 2.3. Deconstructing Behavior Dynamics by Active Projection

We sought to know further details about how the performance metrics evolve with iteration. To address this question, we applied the Active Projection method recently proposed [29] (see Methods). This enabled us to track a model’s dynamic balance between its True Positive Rate (TPR, also called sensitivity) and True Negative Rate (TNR, also called specificity). Notably, the method visually clarifies why TPR or TNR can be high and yet MCC or F1 can be low for a model.

Active projections on the external probe dataset using curiosity-picked CATS2D-dipeptide and ECFP-dipeptide models are shown in Figure 3. They clarify that predictions for initial iterations of the CATS2D-dipeptide model were heavily biased toward inactive predictions, as the TNR was close to 0.75 but the TPR rate was less than 0.25; the resulting MCC was approximately 0 (Figure 2, left panels). However, the curiosity-picked examples iteratively contributed to a more balanced model, which can be seen from the active projection. Where the CATS2D-dipeptide model reaches its approximate peak of MCC = 0.50, the TPR and TNR were 0.75 and 0.80, respectively.

Interestingly, the corresponding ECFP-dipeptide model followed a different trajectory, with lower initial TNR, higher initial TPR, and an initial evolution that better predicted the inhibiting ligand-target pairs. At approximately 4% of the training data, the model has uncovered the patterns in the bioactive non-probe data with matching correspondence in the probe dataset, and further data selection from the non-probe data enhances selectivity prediction in the probe dataset. The model finally arrives in a stable zone around TNR 0.85, TPR 0.70 and MCC of 0.50 at 20% of training data (Figure 3, right panels).

Analyses with pChem-dipeptide and pChem-tripeptide displayed jittering behavior in early iterations of learning. For instance, by use of active projection, we observed the dipeptide-based model had a high TNR before the model shifted focus to actives, and subsequently developed predictive ability of both actives and inactives (Appendix A). For the tripeptide-based model, TPR was high early on with subsequent shift to inactives. Eventually both models stabilized at TPR of more than 0.80 and TNR of more than 0.65 with MCCs near 0.50. The dynamic behaviors of active learning were clearly deconvoluted through the use of active projection.

### 2.4. Feature Analysis and Sequential Decision Making from Features

In chemogenomic active learning, the model is always trying to establish statistical patterns correlating chemical-protein descriptors to the endpoint (active or inactive in this case). Having identified that target descriptors did contribute to probe compound profile prediction, and having deconstructed the evolution of model performance by active projection, we also wanted to understand which of the descriptors contributed to this pair of results. To find out the answer, the time series evolution of feature weights was calculated and visualized by heatmap representation (see Methods).

Feature weights for the CATS2D-identity and CATS2D-dipeptide experiments, for up to 20% of training on the non-probe data, are shown in Figure 3. It is essential to note that because weights differ in the experiments and the relative scalings of feature weights are different, only intra-feature comparison in a single experimental setting (i.e., fixed descriptors) is appropriate. For intra-experiment comparison, we can only compare the values of weights relative to the average weight per iteration. The feature weight heatmaps of CATS2D-identity and CATS2D-dipeptide experiments show that some of the features from CATS2D descriptors are highly weighted from early iterations (e.g., hydrogen bond donor atom pairs or donor-lipophilic atom pairs). Some of the CATS2D descriptors are almost never considered (e.g., positively/negatively-charged atom pair), either by coincidence due to random feature selection for evaluation, or due to lack of discriminative power. Other descriptors are initially uninformative or not considered yet are factored into models at later iterations. This pattern was present in both identity- and dipeptide-described experiments (Figure 4). As a result of scaling, analysis of the CATS2D-dipeptide feature weights demonstrates that target descriptors were indeed contributing to decisions made by the random forest, albeit with lower frequency than the compound descriptors (Figure 4).

Examination of the target descriptor weights in the CATS2D-identity experiment reveals that one of the identity descriptors had unusually high weighting. This would suggest that chemogenomic active learning is building per-target SAR models within the structure of the decision trees developed by placing target identification decision nodes at early decision branches, a hypothesis considered previously in the context of a much larger ligand-target dataset [15]. This hypothesis is explored in further detail below.

A similar analysis was performed for the pChem-identity and pChem-dipeptide models (Appendix A). Most of the highly weighted features belonged to the physicochemical descriptors. Among the chemical descriptors, some of the features are always weighted low, i.e., the number of sp hybridized carbons, total charge, the numbers of 11- and 12-membered rings, and the distance/detour ring indices of order 6, 11, and 12. On the other hand, some of the features including the percentages of hydrogen, carbon, nitrogen and oxygen, total polarizable surface area, logP estimates, compound surface areas (total, acceptor, or donor), hydrophilic factor, Ghose-Crippen molecular refractivity, McGowan volume, and van der Waals volume from McGowan volume, are often weighed highly (Appendix A). Compound feature weightings were similar for the identity- and dipeptide-type models. Via feature weight analysis, it is clear that dipeptide frequencies were also factors in model behavior, though iteration-dependent feature weight evolution knowledge was difficult to extract.

Considering reproducible improvement from target descriptors (Figure 2 and Table 2) and that target feature values were part of decision tree formulation (Figure 4), we wanted to even further clarify the behavior of the system. Toward this goal, we considered deciphering the decision making process through visualization of decision trees.

Representative decision trees from the chemogenomic random forest, where decision trees were fit at the 10% data level with curiosity picking, were rendered (Figure 5). For comparison to Figure 3 and Figure 4, the CATS2D-dipeptide descriptor was again used. Complementary to Figure 4, the order in which the features are considered is made unambiguous. To our knowledge, this is the first report of a chemogenomic model that unambiguously details the decision making process by a statistical pattern recognition (machine learning) estimator.

Consistent with Figure 4, it is evident from this visualization that the majority of the decision nodes use the chemical descriptors. We see from Figure 5 that pharmacophoric features, notably those groups labeled in Figure 4, are highly present in early decision making nodes. While CATS2D considers all pairs of atom types separated by 1–10 bonds (0–9 intermediate atoms), and we see high weighting on all path lengths for some of the pharmacophoric groups (Figure 4), short-distance atom pairs appear more frequent in early decision nodes. Nonetheless, it is clear that peptide subsequence information leads to both local SAR development as well as to final active/inactive decisions. By comparing with Table 2, it is notable that the individual trees in Figure 5 have performance approximate to or only marginally less than the forests that they belong to.

### 2.5. Consequences of Subsequence Absence-Presence Tests; Local SAR Model Development

Decision tree visualization contributed to a clarification that peptide sequences were in fact a part of the active/inactive decision process; comparing them to trees with identity descriptors confirmed their higher relative weights (frequency of use in decision making, data not shown). Yet, it is not intuitive from a biological standpoint how a single dipeptide or tripeptide without any other structural or biophysical context could explain activity. Spurred by a desire to unambiguously explain machine learning, an additional in-depth analysis was executed.

First, a multiple sequence alignment of the MMPs was performed using the Clustal Omega service [30]. Decision tree branches were visually inspected for dipeptide and tripeptide sequences, then compared to the multiple sequence alignment. This revealed that some tripeptide sequences were unique to a single MMP (e.g., the tripeptide “MSL” was only present in MMP9, see Figure 5). In some cases, (in)active decision was an immediate consequence of the subsequence presence test. In other cases, a collection of chemical descriptors was cascaded below the tripeptide until a final decision was made. This suggests that the tripeptide sequences are simply markers that provide fast access to the identification of a target and its SAR. Based on manual analysis of rendered trees, we have compiled a rough sketch of tripeptide-based decisions possible, available as Appendix A.

Reducing the resolution to dipeptides also demonstrated this effect, but decision nodes tended to use dipeptide count thresholds higher than one. Nonetheless, cascades of dipeptide decision nodes or alternating compound-protein nodes made clear that target identification with subsequent SAR generation was the dominant process for final classification of ligand-target interaction.

It is also worth considering that there were loose correlations in training data between MMP targets and their fraction of active data points. For example, the Pearson correlation between the fraction of actives per target and asparagine-valine (NV) dipeptide frequency was 0.43. Hence, despite lacking biophysical context, such a dipeptide fills the role of a first-line inquiry (Appendix A). After substantial manual inspection and discovery of such correlations, we proceeded to systematically quantify the percentages of decision trees that had target descriptors as their root decision node. Compared to the frequency of dipeptide descriptors as root decision nodes (95% confidence interval: 20–28% of decision trees per n = 5 random forests), tripeptides were significantly more present as roots (95% CI: 38–54%).

As mentioned above, all we could do after initial experiments (Figure 2) was to speculate about correlations between descriptors and active/inactive relationships. However, combining those results with the feature weighting and detailed tree rendering (Figure 5) has yielded a far clearer picture about how ligand-target interaction decisions are made, and how we should design computational experiments for interaction modeling.

### 2.6. Yoked Prediction by SVM and ANNs

In experiments so far, we have only trained random forest models. We also wanted to expand prospective evaluation to other machine learning algorithms such as SVMs and ANNs by feeding them with data points selected by active learning. At least for ANNs, the idea to simply swap out one algorithm with a replacement ANN is also referred to as “shallow deep learning” [31]. In experiments with CATS2D-dipeptide descriptors, SVMs and ANNs were able to achieve relatively high maximum MCC values between 0.60 to 0.70 in several experimental settings. At first glance, these algorithms can be said to outperform the random forest models. Yet the metrics were generally unstable and small changes to the model configuration such as the number of hidden layers or regularization coefficient easily resulted in a model with much lower MCC of around 0.40, or a dysfunctional model which classifies all data points into one class and does not have measurable MCC (data not shown).

By comparing ANN models with a different number of layers, it was evident that adding more layers to ANN models did not significantly improve the MCC. Although the highest MCC of 0.72 was achieved with an ANN having 10 layers, MCC of approximately 0.70 could also be achieved by an ANN having six or seven layers. Even smaller, we found that an ANN with two hidden layers could achieve performance of MCC = 0.71 on the external set (75% of training data, regularization alpha = 0.1), yet the very same architecture and regularization could only achieve MCC = 0.46 when initialized using a different seed value. SVM models yielded identical prediction performance across multiple trials with different seeds, as the SVM algorithm is guaranteed to converge on a unique, optimum solution.

We should also note that some individual decision trees trained on 75% of the training data could achieve MCC of 0.72. In both RF and ANN, random number generator seed values affect the unitary decision elements of the estimators (respectively, decision trees and non-input units of ANNs). While the standard RF algorithm does not feature a feedback correction mechanism similar to backpropagation in ANNs, it would appear that weight initialization in ANNs is sensitive to the point that even application of backpropagation cannot adjust faulty-initialized decision criteria in network units enough to recover the network to a state of predictive utility. Since there is no prediction difference in an optimal decision tree and a full ANN model, we would also be led to believe that ANNs, as constructed here using the multi-layer perception implementation, are similarly performing the target identification and local SAR modeling steps by adjustment of (linear) perceptrons that are cascaded together for final decision making.

## 3. Materials and Methods

### 3.1. Ligand-Target Activity Retrieval and Discretization; Probe Compound Selection

MMP ligand-target bioactivity was obtained from the ChEMBL database, version 24, using the publicly available Python-language programming interface (data retrieval February 2019). Inhibition constant (*K_i_*) values were retrieved, and 10 MMP targets with at least 50 compounds per MMP target were retained. Per-target, mean or weaker *K_i_* was classified as inactive, and mean + 30-fold or stronger *K_i_* was classified as active. Targets with mean *K_i_* values stronger than 1500 nM were re-adjusted to 1500 nM in order to avoid the boundary of the active class from becoming impractical (e.g., considering compounds with 40 nM activity to not be classified as actives). Thus, the limit of actives was no stronger than 50 nM. By systematic auto-discretization, all MMP targets had 1500 nM/50 nM limits with one exception (MMP7, 1803 nM/60 nM). Post-discretization, targets were retained if at least 30 compounds remained; MMP1-3, MMP7-9, and MMP12-14 were retained for experiments (e.g., MMP10 had 59 compounds with annotation but 45 had *K_i_* values inside the range 1500 nM-50 nM; MMP10 was thus removed). Probes were defined as compounds with an active annotation on exactly one target and an inactive annotation on one or more other targets (Figure 1). Probes and non-probes were isolated for respective use as external prediction and training datasets.

### 3.2. Compound and Target Descriptor Computation

Compounds were described by three representations. First, real-valued physicochemical properties were computed using the commercial DRAGON 7 package (Talete s.r.l., Milano, Italy; “Constitutional”, “Ring Descriptor”, and “Molecular Properties” blocks, respectively comprising 47, 32, and 20 descriptors). Three toxicity estimate descriptors were removed, yielding a total of 96 descriptors for modeling. Second, the DRAGON implementation of the CATS1 pharmacophore representation of compounds [32] was computed, yielding 150 non-negative integer-valued descriptors. Third, the OpenEye OEGraphSim library was used to compute atom-centered circular fingerprints with a maximum radius of two; binary fingerprints were hashed into 512-, 1024-, and 4096-bit representations to investigate any effects from hash collisions on model performance.

Targets were described by two representations. First, a dummy identity vector unique to each target was generated for use in control experiments; hence the collection of dummy vectors forms an identity matrix. Second, protein primary sequence subsequence frequencies (“k-mers”) were tabulated for lengths one, two, and three [33]. For multimer frequency calculation, the overlapped sliding window approach was used (e.g., for sequence MKSLP in MMP3, computation of a 2-mer is a vector of length 4 and a 3-mer is of length 3 comprised of subsequences MKS, KSL, SLP each occurring once).

### 3.3. Active Learning Implementation

Chemogenomic active learning using random forest as the underlying estimator algorithm was employed similar to previous studies [15,16,34]. Based on previous studies [16,28], the number of trees to comprise a forest was set to 100. Models were iteratively updated by selecting one ligand-target pair at a time from the non-probe data, per selection strategy [15]. At each model update, predictive performance on the external probe dataset was evaluated by MCC and F1 metrics (see below). Active learning on the non-probe training data was carried out until data was exhausted. Experiments were repeated 10 times to minimize the impact of randomization effects, as the random forest algorithm applies both random selection of examples (bagging) and iterative random selection of descriptors to consider for discriminative ability. Both types of random selection are dependent on a seed value used to initialize the system’s random number generator. The method begins with a randomly picked active and inactive annotation, which is also seed-dependent.

### 3.4. Performance Metrics

Prediction results could be one of four types: true positives (TP), which are correct inhibition/active predictions; true negatives (TN), which are correct non-inhibition/inactive predictions; false positives (FP), which are predictions of inhibition that are actually inactive after discretization, and false negatives (FN), which are actives predicted to be non-inhibition. Using these four types of results, MCC and F1 can be computed respectively by the following formulas:MCC = (TP × TN − FP × FN)/sqrt((TP + FP) × (TP + FN) × (TN + FP) × (TN + FN))F1 = (2 × PPV × TPR)/(PPV + TPR)(1)
where PPV and TPR are the positive predictive value and true positive rate, defined as:PPV = TP/(TP + FP) TPR = TP/(TP + FN)(2)
During computation of metric surfaces [35] and active projections [29] the true negative rate (TNR) is also computed:TNR = TN/(TN + FP)(3)

### 3.5. Active Projection Implementation

Active projection [29] analyzes the evolution of prediction performance with respect to a primary metric such as MCC by overlaying the iterative change in TPR and TNR on top of the metric surface [35] computed from the ratio of actives to inactives. An in-house implementation was employed.

### 3.6. Feature Weight Matrices

The active learning cycle was amended to output the weights of features as determined by the random forest algorithm implementation in the scikit-learn package [36]. In short, features are weighted proportional to the number of times they are used by decision tree nodes. At specified intervals of active learning, feature weights are recorded. The per-iteration collection of weights is aggregated into a matrix, and raw feature weights are transformed per iteration by subtraction from the mean and division by standard deviation (“z-scaling”), yielding relative feature weights. Feature-weight matrices are visualized by in-house tools.

### 3.7. Decision Tree Visualization

Decision rules of individual trees were visualized by customizations of tree export functionality available in scikit-learn (version 0.20.1, installed December 2018) [36]. Trees are exported to the open source DOT format and subsequently rendered.

### 3.8. Yoked SVM/ANN Experiments

By recording the sequence of ligand-target pairs, models can be later reconstructed. However, in addition to chemogenomic random forest models, it is also possible to use other estimator algorithms such as SVMs and ANNs. We used the scikit-learn implementation of both algorithms (ANN: multi-layer perceptron) to test if models derived using the curiosity picking histories could results in better performances. Whereas the random forests were built with a single parameter (number of trees), SVMs required iteration of a grid of tolerance (loss, C) and radial basis function kernel parameter (gamma) settings. Linear SVMs were also tested. For ANNs, the number of features in CATS2D-dipeptide models, 537 descriptors, was used as the size for all hidden layers, and feedforward networks consisting of one to ten hidden layers were systematically tested. For each network topology, six regularization values (alpha/tolerance) crossed with five random seed values were used to compute ANN models, which were subsequently tested on the external probe dataset.

## 4. Conclusions

Chemogenomic active learning has emerged in the past few years as a highly attractive strategy for productivity gain in drug discovery. Yet it has been unclear how it would perform in a prospective setting where compound profiles to predict are those similar to chemical probes and thus have little active/positive annotations. We challenged the algorithm in this setting and found that it still exhibited reasonable (TPR > 0.70, TNR > 0.70) performance. More surprisingly, we uncovered that target descriptors can in fact make a substantial improvement in models, something that was unclear in previous studies [15].

The active projection method was utilized to explain how the model evolved in terms of TPR and TNR rates, and provided the rationale for interpreting MCC/F1 values. It also provided a visual analysis of how the model moves in TPR/TNR space when it appears that maximum predictive ability (MCC/F1) is stalled despite adding more data to models.

Feature weight analyses and unambiguous decision tree rendering made clear how random forests were arriving at decisions, and the order in which decisions were made. The extensive follow-up analysis revealed the mechanics of how machine learning employs the target descriptors to arrive at more performant models. As Bajorath and colleagues have already previously suggested that predictions are dominated by nearest neighbor effects [7], and since Rakers et al. replicated such a proposition by demonstrating the divide between success and failure in simulated orphan bioactivity profile prediction [16], we would conclude that the choice between single target SAR modeling and chemogenomic modeling will largely be decided on the availability of bioactivity data for the target in question and the target’s homology to the family it belongs to. With little to no data for a target, chemogenomic modeling can be expected to create a bridge to the uncovering the target’s SAR.

We found replacing random forests with SVMs and deeply-layered ANNs could potentially improve prospective prediction MCC, yet it was highly concerning to see that an identical network architecture could yield high predictive performance with one random seeding and completely useless performance with another random seeding. This underscores the commentary by some researchers that considerable caution is needed with neural networks [37]. These results re-iterate the importance of employing an ensemble of multiple models for prospective projects. Schneider has previously proposed the “jury network” for combining and weighting the results of multiple models toward reliable decision making [38].

Taken in total, this work represents a positive step forward in techniques and expectations for computer-aided molecule design toward the design and discovery of chemical probes. Given the low number of compounds simultaneously satisfying selectivity, potency, and permeability requirements [2], there is much work to be done in probe development.

## Figures and Tables

**Figure 1 molecules-24-02716-f001:**
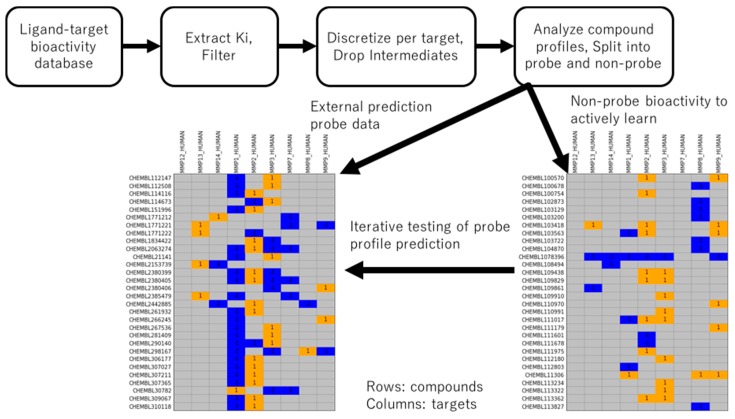
Extraction of bioactivity data and division into probe and non-probe datasets. MMP Ki bioactivity is extracted, filtered to ligand-target pairs with consistent measurement, discretized by a 30-fold difference in activity per target, and compounds are classified as probes or non-probes based on remaining data. Probes contain exactly one active annotation and at least one non-active annotation.

**Figure 2 molecules-24-02716-f002:**
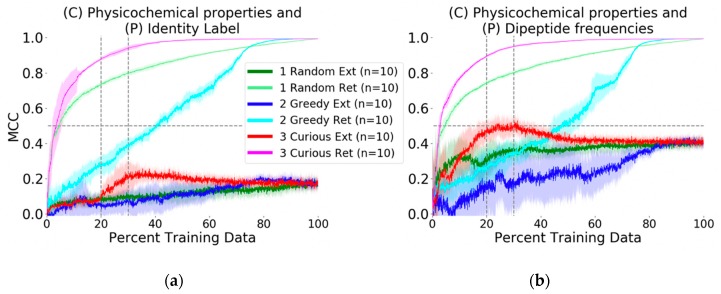
Longitudinal evaluation of active learning. Results are grouped and colored by picking methodology, with light tones for retrospective (“Ret”) evaluation and dark tones for external (“Ext”) dataset evaluation. Curiosity-driven active learning on the MMP non-probe training dataset demonstrates high retrospective predictive ability against the full training set using only 20% of the available training data. However, when targets are described by simple identity vectors (control experiment, panel (**a**)), prediction of external probe compound profiles is poor, regardless of ligand-target pair picking strategy. In contrast, describing targets by overlapping primary subsequence frequency yields information that substantially improves the random forest’s ability to discriminate active ligand-target pairs from inactive ones in the external data (panel (**b**)).

**Figure 3 molecules-24-02716-f003:**
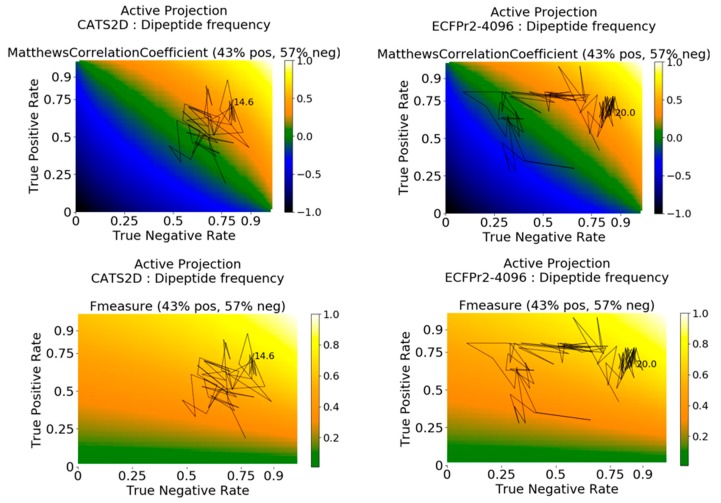
Active projections decompose the evolution of active learning into active and inactive prediction rates in an iteration-dependent fashion, in addition to the primary metric. Here, the active projection of probe molecule profile performance is shown for the MCC (**top**) and F1 (**bottom**) metrics. The ratio of active to inactive annotations is used in generating metric surfaces for each type of metric used, guiding interpretation about the range of possible values under such ratio. Active projections terminate when no substantial change (here: 0.075) in either TPR or TNR can be achieved, and the final amount of training data at projection termination is given.

**Figure 4 molecules-24-02716-f004:**
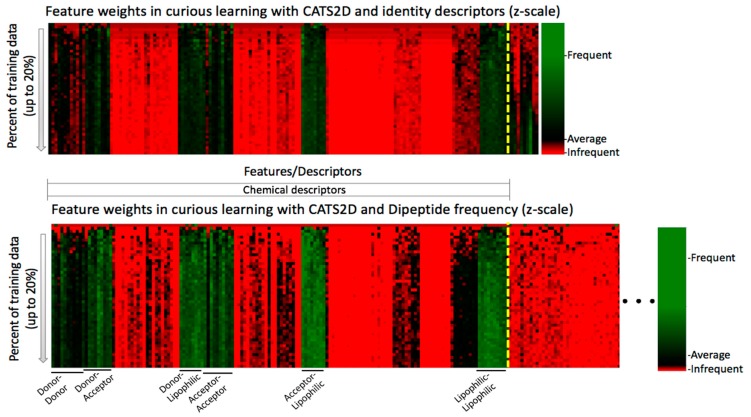
Feature weight time series show the evolving relative weights of each descriptor used, oriented from **top** (initial iterations) to **bottom** (later iterations). A yellow dotted line denotes the boundary between chemical and protein descriptors. Order of compound descriptors is preserved between top and bottom panels. Protein dipeptide frequencies are less weighted yet still non-trivial. The pharmacophoric patterns that most influence decision making are labeled. (Bottom panel—truncated figure. Original, full heatmap can be found in Appendix A.).

**Figure 5 molecules-24-02716-f005:**
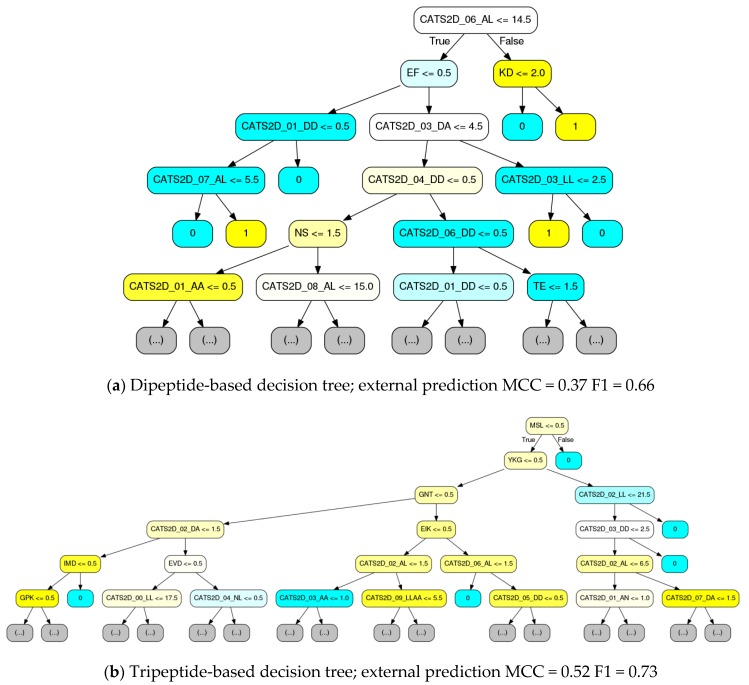
Deconvolution of ligand-target interaction prediction by custom visual rendering of decision trees (DTs) comprising random forests. The visualizations provide pinpoint description of the criteria identified in the samples and features presented to the DT-building algorithm (samples and features provided by random selection with replacement, known as bagging). Here, trees were constructed using CATS2D ligand descriptors with (**a**) dipeptide or (**b**) tripeptide descriptors. Node color intensity indicates the amount of purity obtained in for a decision based. Yellow coloring denotes inhibitory activity, and cyan color denotes non-inhibitory activity. Individual MCC/F1 performance on the external set is show below each tree.

**Table 1 molecules-24-02716-t001:** Sizes of MMP non-probe (training) and probe (external prediction) datasets used in the study.

**Actively Learned Training Data: Non-Probe Profiles**			
	**Total**	**Active**	**Inactive**
Annotations	2397	1586 (66%)	811(33%)
Compounds	1181	750	473
Targets	9	9	9
**External Prediction Data: Probe Profiles**			
	**Total**	**Active**	**Inactive**
Annotations	165	72 (43%)	93 (56%)
Compounds	72	72	72
Targets	8	7	8

**Table 2 molecules-24-02716-t002:** MMP non-probe training, probe prediction performance by curiosity-type chemogenomic active learning. Contributions of overlapping target subsequences toward probe profile prediction are confirmed across multiple representations of compounds. Identification of the probe compound interactions is successful as early as 20%, as measured by F1 values. MCC differs from F1 values by further factoring in model discrimination ability with respect to selectivity. In no case does learning the full training set contribute to improved external prediction performance.

Descriptors (Ligand:Target)	Mean MCC at 20% Training	Max Mean MCC (% Training Data)	Mean F1 at 20% Training	Max Mean F1 (% Training Data)
pChem:identity	0.11	0.26 (35)	0.58	0.63 (34)
pChem:residue	0.35	0.44 (34)	0.67	0.71 (29)
pChem:dipeptide	0.46	0.53 (31)	0.72	0.75 (31)
pChem:tripeptide	0.48	0.55 (32)	0.73	0.76 (32)
ECFPr2-1024:identity	0.23	0.26 (22)	0.56	0.58 (21)
ECFPr2-1024:dipeptide	0.59	0.67 (57)	0.77	0.82 (57)
ECFPr2-512:dipeptide	0.62	0.69 (31)	0.78	0.83 (39)
ECFPr2-4096:dipeptide	0.66	0.75 (64)	0.80	0.86 (64)
pChem + ECFPr2-1024:dipeptide	0.57	0.62 (21)	0.76	0.79 (21)
CATS2D(D7):identity	0.21	0.23 (17)	0.53	0.55 (2)
CATS2D(D7):dipeptide	0.53	0.56 (25)	0.72	0.74 (23)
CATS2D(D7):tripeptide	0.56	0.63 (33)	0.75	0.79 (33)
CATS2D(D7):tetrapeptide	0.59	0.64 (32)	0.77	0.80 (32)

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
