# Peer review of "Applicability Domain of Active Learning in Chemical Probe Identification: Convergence in Learning from Non-Specific Compounds and Decision Rule Clarification"

_molecules, 2019, doi:10.3390/molecules24152716_

Round 1
Reviewer 1 Report
This manuscript is a clear description on the use of physiochemical properties to predict ligand-target binding. The results are clear but there is a lack of description for the ANN - for example, how many input nodes and the list of input parameters, what is the ANN architecture?
Author Response
Dear Reviewer,
Thank you for your limited time in reviewing our manuscript.
> The results are clear but
there is a lack of description for the ANN -
> for example, how many input nodes and the list of input parameters, what is the ANN architecture?
The ANN architectures and input was listed in the Methods and Materials section in the original manuscript. We used the CATS2D-dipeptide representation for a RF-ANN-SVM comparison experiment, with 150+387=537 descriptors (387 dipeptides present in at least one MMP) to feed the ANN input layer. Hidden layers were all 537 descriptors, and we tested 2-10 hidden layers (in feedforward fashion).
For your easy confirmation, this section has been highlighted in blue in the revised manuscript (p.15).
Reviewer 2 Report
The authors used active learning approach with Random Forest method to predict interaction between ligand with MMPs. The result show performances over different features used, such as identity, amino acid frequency, dipeptide frequency etc.
The result sound reasonable, but without much insight, especial relation to biological and chemical problem. The paper is merely a description about an application of the method.
With dipeptide frequency as an example, the difference in dipeptide frequency is a superficial part to the difference in protein sequence that barely be mentioned. A comparison between the 9 sequence,multiple sequence alignment by clustalw or other program would help to highlight the biological meaning and understand Figure 4 .
The same applies to the chemical description of ligand.
Minor,
a. 4.2. Compound and target descriptor computation
in "computation of a 2-mer is a vector of length 4 and a 3-mer is of length 3", is "a vector of length 4" instead of a vector of length 2, for speeding up or essential? if the later, why?
b. Figure 1. could be improved for clarification. larger legend and axis label, and the order of legend, the text of of legend.
c. could some features be manually labeled on x-axis of Figure 3?
Author Response
Dear reviewer,
Thank you for contributing your limited time to a thorough review of our work.
We are highly appreciative of your critiques to address the results in a more biologically meaningful way.
Thanks to your comments, we have performed extensive additional experiments and gained significant new perspectives.
The major changes to the manuscript are colored in red for your quick confirmation.
A point-by-point response to your critiques is as follows.
[1] With
dipeptide frequency as an example, the difference in dipeptide frequency
is a superficial part to the difference in protein sequence that barely
be mentioned. A comparison between the 9 sequence,multiple sequence
alignment by clustalw or other program would help to highlight the biological meaning and understand Figure 4
Thank you for this suggestion to perform a multiple sequence alignment.
We dutifully performed the alignment of the nine MMP targets and cross-checked the individual decision tree rules with this alignment.
Searching the alignment for the dipeptide and tripeptide frequencies found at early decision nodes revealed that some tripeptides were unique to individual MMP targets.
Considering that we had already discussed the absence of biological and chemical context in machine learning algorithms in the original manuscript, we interpreted these new results as reinforcement of that discussion, suggesting that the improvement by non-identity descriptor is because it more rapidly identifies the target (by increased opportunity from target descriptor length and unique subsequence) and then proceeds to build an efficient local SAR model.
We then analyzed the targets by their active-to-inactive ratio and such correlation with a few manually selected protein subsequences, finding loose correlation. That also matched the fact that some protein descriptor decision nodes led directly to active/inactive decisions.
We followed that up by a small analysis looking at whether target or compound descriptors dominate the root nodes of decision trees, finding that tripeptides were significantly more common than dipeptides in root nodes (demonstrated by confidence intervals).
Therefore, for applied chemical biology and drug discovery projects, target similarity and similar target SAR availability will dominate the choice of chemogenomic or local SAR modeling.
All of these results and new viewpoints have been incorporated into the main manuscript, mostly in new Section 2.5 beginning on page 11 of the revised manuscript.
We hope you will find them to be a substantial improvement to the manuscript.
[2] a. 4.2. Compound and target descriptor computation in "computation of a 2-mer is a vector of length 4 and a 3-mer is of length 3", is "a vector of length 4" instead of a vector of length 2, for speeding up or essential? if the later, why?
These vector lengths are a direct consequence of using the sliding window count approach, and not related to computational speed.
We have provided a concrete example of the 3-mer calculation in the revised manuscript (p.14, Section 4.2).
[3] Figure 1. could be improved for clarification. larger legend and axis label, and the order of legend, the text of of legend.
Thank you for the critique. Figure 1 (now Figure 2 in the revised manuscript) was updated to use larger fonts in all aspects. The redundant legends were replaced with a single legend. The order of the legend is intentionally set to show the control experiment (random picking) first, followed by greedy and curiosity performance. We have explicitly noted in the legend that results from each picker are grouped by retrospective-external prediction using a similar color for the pair of results. The updated legend and caption can be found on page 5 of the revised manuscript.
[4] c. could some features be manually labeled on x-axis of Figure 3?
An excellent point. We have re-generated results with each feature labeled in the time-series feature weight matrix. However it is too difficult to see individual labels when the figure is resized to standard A4 paper format. We instead have annotated, in the main manuscript, blocks of related pharmacophoric descriptors that were (relatively) higher weighted. The revised figure and caption can be found on page 9 (now Figure 4) of the revised manuscript.
Reviewer 3 Report
This is a high quality study in which the concept of 'active learning' is applied to the problem of chemical probe identification. It is well written but requires a number of reading cycles before the methodology and results become clear. For this reason, I would like to propose a number of changes that might improve the readability of the paper:
Clearly define what is meant by a 'probe' (it is defined on page 2, but adding an additional paragraph on this would make it more readable).
Table 1, and the paragraph describing this table in the text: this is not very well explained and I would suggest to elaborate further on this so that it becomes more clear to the reader. Maybe it would be useful to add a figure as well.
Figure 1 is of low quality and very hard to interpret. Please describe in the figure caption the differences between the different curves (i.e. explain the labels in the legend).
Table 2: given the variations in the calculated MCC and F1 values, the number of significant digits in this table seems not appropriate.
Explain why the MCC and F1 metrics were used in this work, and not any other metric like for example the accuracy, precision or power metric (Lopes et al, J. Cheminformatics 2017, 9:7).
Figure 4: although this figure is useful to illustrate some of the conclusions of the work, it is nevertheless very hard to interpret and not very informing for those readers who are interested in the general conclusions of the work.
Materials and Methods section:
Should the 'upper limit of inactivity' not read as 'lower limit of inactivity', and 'lower limit of activity' as 'upper limit of activity'?
Explain in more detail what is meant with a 'dummy ideate vector' to describe each target. Is this simply the name of each target?
Page 3 and 4: although the concept of curiosity-, greedy- and random-picking is explained in reference 12, it would nevertheless clarify the paper and improve readability of a small explanation of these terms is added to the Materials and Methods section.
Author Response
Dear reviewer,
Thank you for contributing your limited time to a thorough review of our work.
We are most encouraged by your favorable evaluation of our study, and highly appreciative of your critiques to make the paper even more impactful.
Thanks to your comments and those of other reviewers, we have performed extensive additional experiments and gained significant new perspectives.
The major changes to the manuscript are colored in red for your quick confirmation.
We believe you will find this manuscript to be of even higher quality.
A point-by-point response to your comments is as follows.
[A] Clearly define what is meant by a 'probe' (it is defined on page 2, but adding an additional paragraph on this would make it more readable).
--> Thank you for this critique. We have amended our manuscript in two ways.
First, we added an explicit sentence to the Results and Discussion subsection (2.1) that probes contain strong potency for exactly one target and non-potency for at least one other target (p.3 in the revised manuscript).
Second, we have created new Figure 1 which explicitly shows the workflow and a subsample of the ligand-target matrix containing active/inactive annotations, where probe and non-probe profiles are split into separate matrix visualizations. This makes it visually clear what is meant by probe and non-probe profiles.
[B] Table 1, and the paragraph describing this table in the text:
--> As above, new Figure 1 should help to understand what is meant by probe and non-probe, and further details about exclusively-active compounds and exclusively-inactive compounds.
[C] Figure 1 is of low quality and very hard to interpret. Please describe in the figure caption the differences between the different curves (i.e. explain the labels in the legend).
--> Thank you for the critique. Figure 1 (now Figure 2 in the revised manuscript) was updated to use larger fonts in all aspects. The redundant legends were replaced with a single legend. The thickness of line colors was expanded to help more rapidly identify experiment types. The order of the legend is intentionally set to show the control experiment (random picking) first, followed by greedy and curiosity performance. We have explicitly noted in the legend that results from each picker are grouped by retrospective-external prediction using a similar color for the pair of results. The updated legend and caption can be found on page 5 of the revised manuscript.
[D] Table 2: given the variations in the calculated MCC and F1 values, the number of significant digits in this table seems not appropriate.
--> Thank you for the critique. Indeed, significant digits were inconsistent in the original manuscript; we apologize for the issue.
In the revised manuscript, all MCC/F1 metrics were reported to two significant digits, and all percentages were rounded to integer quantities.
[E]
Explain why the MCC and F1 metrics were used in this work, and not any other metric like for example the accuracy, precision or power metric (Lopes et al, J. Cheminformatics 2017, 9:7).
--> Thank you for the reference to the recently-proposed Power Metric. We were unaware of this new metric. We implemented the metric and analyzed its character by the Metric Surface and iCDF analysis methods (Brown 2018 Mol Info). While the metric is indeed data balance-independent, we found there was potential for it to over-estimate performance, much like Accuracy. In re-evaluation of our results using the Power Metric, we found it was clearly higher than MCC, and in some cases higher than Accuracy. We also found a Commentary article in the same journal that criticized the metric in a similar way. Therefore, we discussed the Power Metric and Accuracy metrics in the revised main manuscript, but did not change the presentation of results in Table 2 to include either metric. The reference to PM and the critique article are new references 25 and 26 (updated discussion on page 5 of the revised manuscript).
[F] Figure 4: although this figure is
useful to illustrate some of the conclusions of the work, it is
nevertheless very hard to interpret
--> We agree that it was a challenge to interpret such a large decision tree. Spurred by the comments of reviewer #2, we sought to make further interpretation of machine learning based on manual analysis of the original Figure 4.
We performed an alignment of the nine MMP targets and cross-checked individual decision tree rules with this alignment. Searching the alignment for the dipeptide and tripeptide frequencies found at early decision nodes revealed that some tripeptides were unique to individual MMP targets.
Considering that we had already discussed the absence of biological and chemical context in machine learning algorithms in the original manuscript, we interpreted these new results as reinforcement of that discussion, suggesting that the improvement by non-identity descriptor is because it more rapidly identifies the target (by increased opportunity from target descriptor length and unique subsequence) and then proceeds to build an efficient local SAR model.
We then analyzed the targets by their active-to-inactive ratio and correlation with a few manually selected protein subsequences, finding loose correlation. That also matched the fact that some protein descriptor decision nodes led directly to active/inactive decisions.
We followed that up by a small analysis looking at whether target or compound descriptors dominate the root nodes of decision trees, finding that tripeptides were significantly more common than dipeptides in root nodes (statistically tested by confidence intervals).
Therefore, for applied chemical biology and drug discovery projects, target similarity and similar target SAR availability will dominate the choice of chemogenomic or local SAR modeling.
All of these results and new viewpoints have been incorporated into the main manuscript, mostly in new Section 2.5 beginning on page 11 of the revised manuscript.
[G] Should the 'upper limit of inactivity' not read as 'lower limit of inactivity', and 'lower limit of activity' as 'upper limit of activity'?
--> We struggled with choice of expression when writing the original manuscript, and understand your critique very well. In the revised manuscript, we have discarded the use of upper/lower terms, and instead unified description by stronger/weaker terms. We believe the revised manuscript is clearer. The text revisions can be found on page 13 of the revised manuscript.
[H] Explain in more detail what is meant with a 'dummy ideate vector' to describe each target. Is this simply the name of each target?
--> Yes, this is simply the name of each target. We have added a small explanation to Section 2.2 of the main manuscript (page 5 in revision). As an additional experiment to test our revised interpretation of machine learning, we created duplications of the identity matrix which would provide increased probability for subsampling the target identity by the random forest algorithm. These experiments did not improve performance (between one and four duplicates tested). No change in performance was observed, meaning the same low performance on the external probe dataset. These results were _not_ added to the manuscript.
[I] concept of curiosity-, greedy- and random-picking is
explained in reference 12, it would nevertheless clarify the paper and
improve readability of a small explanation
--> Thank you for the suggestion. We have added substantial content to the Introduction section to help the reader unfamiliar with active learning and picking strategies. First, we explained the concept and elements of active learning iteratively picking the data. Second, a paragraph was added immediately afterward to explain the concepts and consequences of each picker. These revised descriptions can be found on pg. 2-3 of the revised manuscript.
Round 2
Reviewer 1 Report
The authors had addressed all of my concerns.